# Few Shot Image Generation Using Conditional Set-Based GANs

## Abstract

While there have been tremendous advances made in few-shot and zero-shot image generation in recent years, one area that remains comparatively underexplored is few-shot generation of images conditioned on sets of unseen images. Existing methods typically condition on a single image only and require strong assumptions about the similarity of the latent distribution of unseen classes relative to training classes. In contrast, we propose SetGAN - a conditional, set-based GAN that learns to generate sets of images conditioned on reference sets from unseen classes. SetGAN can combine information from multiple reference images, as well as generate diverse sets of images which mimic the factors of variation within the reference class. We also identify limitations of existing performance metrics for few-shot image generation, and discuss alternative performance metrics that can mitigate these problems.

## 1 Introduction

Few-shot and zero-shot learning has been an area of exploding interest in machine learning over the past few years. Transformer-based models such as GPT-4 (OpenAI, 2023) have achieved incredible leaps in performance in few-shot text generation, while diffusion-based Ho et al. (2020) image generation models such as DALL-E 3 (Betker et al., 2023) and Stable Diffusion (Rombach et al., 2022) have achieved remarkable success at zero-shot text-to-image generation. One area that remains comparatively underexplored, however, is few-shot or zero-shot image-to-image generation - particularly in the setting of generating images conditioned on *sets* of images.

We propose SetGAN - a novel image generation model that is trained to generate images conditioned on sets of reference images of unseen classes. The model learns to extract relevant features from the unseen reference sets, then generate high-quality, diverse images similar to the reference images at inference time. Once pretrained on a given image dataset, SetGAN can then generate any number of images for a variety of unseen reference classes, all without any further training or finetuning.

Conceptually, this model uses a similar framework to models such as DAGAN (Antoniou et al., 2017), following an adversarial learning approach where a "generator" model attempts to generate images conditioned on a given input image, and a "discriminator" model learns to distinguish between the generated images and other true images from the same class. The difference lies in the set-based nature of our model - SetGAN can condition its generations on multiple reference images rather than just a single image, and similarly generate multiple output images as well. This allows the model to better understand the variations within the reference class, as well as producing diverse sets of output images conditioned on that class. The discriminator is also able to compare the generated sets of images to the reference set, and judge the generated sets based not only on the individual images' similarity to the reference class, but also on the diversity and factors of variation within each set - leading to generations that more closely match the variations within the true reference class.

Existing works frequently rely on learning factors of variation within a typical reference class at training time, then applying those variations to a single image at inference time. This makes the assumption that the factors of variation within a given class at training time will be the same as for the unseen test classes - an assumption that does not always hold. Works that follow this methodology also have a tendency to produce

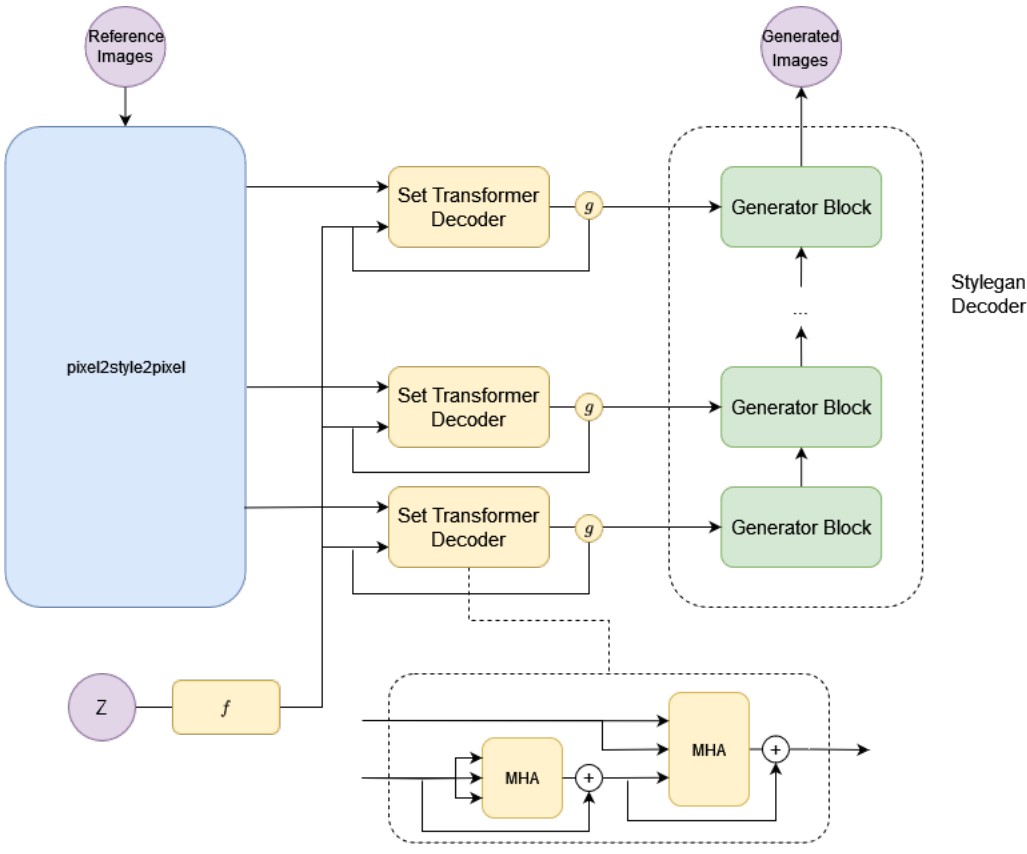

Figure 1: Diagram of the SetGAN generator. The pixel2style2pixel (pSp) encoder maps each input image to the latent space $\mathcal{W}+$. The input style vectors are then passed through the StyleGAN2 mapping network, then passed to a series of conditioning networks which compute conditional styles for each layer of the decoder by attending to the appropriate output layer of the pSp encodings. These conditional styles then become the inputs to the StyleGAN2 generator, which decodes them into images.

generations that are highly similar to the reference image, limiting their diversity. Consider a training set of faces where each class consists of images of the same person under different poses and lighting conditions. If we want to generate faces similar to a reference set that contains images of different women that all have heavy eye shadow, those techniques will generate faces of the same women as the reference set with different poses and lighting conditions instead of different women with heavy eye shadow (see Section 5 and Figure 3). SetGAN does not suffer from these limitations, and can generate truly novel and diverse outputs that reflect the factors of variation of the reference set instead of the training set and without simply reproducing elements of a single input image.

## 2 Related Work

### 2.1 Few-shot GANs

Previous works on few-shot image generation using GANs generally fall into three categories: optimization-based methods, fusion-based methods, and transformation-based methods. Optimization-based methods (Clouâtre & Demers, 2019; Liang et al., 2020) use meta-learning techniques (Finn et al., 2017) to fine-tune their generative models on small amounts of data, but do not produce results competitive with other approaches. Fusion-based methods (Hong et al., 2020b; Gu et al., 2021; Yang et al., 2022) condition on several input images by starting with a single base image and incorporating local features from other reference images. These methods are highly dependent on the images they condition on and sometimes struggle

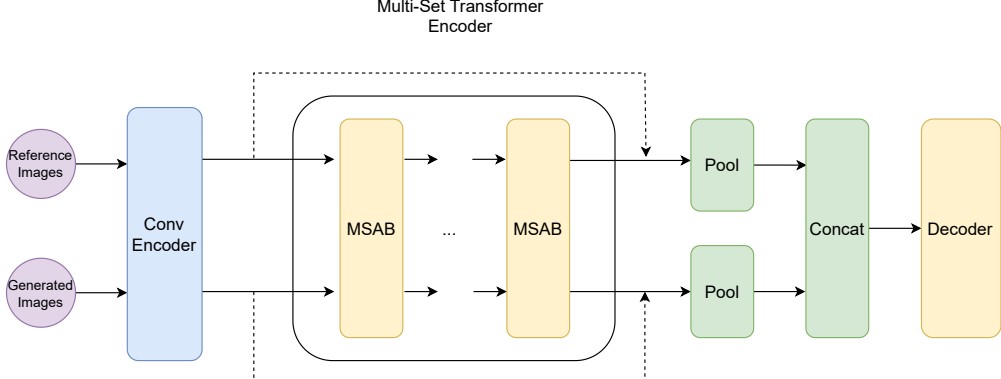

Figure 2: Diagram of the SetGAN discriminator. Sets of input images are encoded as fixed-size vectors using a convolutional network. These sets of vectors are then passed through a Multi-Set Transformer (Selby et al., 2022) consisting of several multi-set attention blocks, followed by a pooling operation performed on each set. These outputs are then concatenated and passed through a feedforward decoder layer to produce a scalar output.

to generalize beyond the features in the input images. Transformation-based methods (Ding et al., 2022; Hong et al., 2020a; Antoniou et al., 2017) learn transformations during training that mimic the typical factors of variation within each training class, then apply these learned transformations to a single test image. These methods can be highly successful at one-shot image generation, but make strong assumptions about the similarity in factors of variation between classes that may not generalize to more diverse datasets. Using only a single image to condition on can also limit diversity, as each generation may be only a slight transformation of the given input image.

## 2.2 Diffusion models

Many diffusion-based approaches such as DALL-E 3 (Betker et al., 2023) and Stable Diffusion (Rombach et al., 2022) have achieved incredible success at text-to-image generation, generating diverse high-resolution images from a wide variety of text-based prompts. These models are largely focused on the domain of text-to-image generation, but some work has been done on the topic of image-to-image generation as well. DreamBooth (Ruiz et al., 2022) and HyperDreamBooth (Ruiz et al., 2023) focus on the task of generating new pictures of a particular subject in new contexts, such as generating pictures of a pet in various locales around the world. More directly, models such as Stable Diffusion Image Variations (Pinkney, 2023) and IPAdapter (Ye et al., 2023) adapt the pretrained Stable Diffusion network to accept image embeddings in order to generate images related to a given image prompt. Similar to other GAN-based methods, these models still accept only a single image as input at a time, rather than a set. Giannone et al. (2022) do propose a framework for few-shot generation with diffusion models conditioned on *multiple* images, however their model is tested only on very low-resolution datasets. As is common for diffusion models, all of these approaches suffer from extremely slow inference speeds - even at lower resolutions.

## 2.3 Image translation

A closely related task to few-shot image generation is image-to-image translation. In this task, the goal is to translate images from one domain to a new domain, often in a few-shot setting. This frequently takes the form of adapting models pretrained on the source domain to the target domain via a minimal number of examples (Li et al., 2020; Ojha et al., 2021). While this approach to few-shot image translation is a distinct task from few-shot image *generation*, some approaches such as FUNIT (Liu et al., 2019b) have combined these approaches by seeking to translate images between different classes of the same dataset.

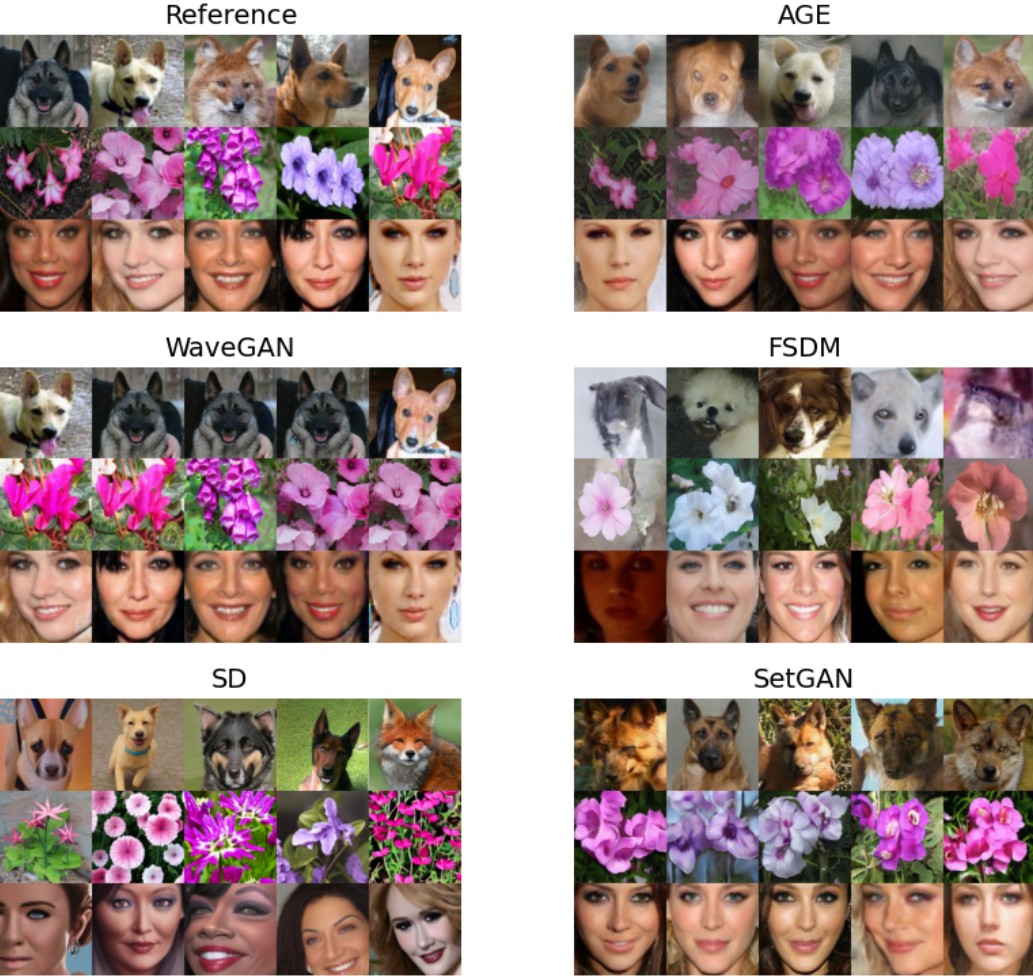

Figure 3: Examples of generations using images across many different test classes that share similarities according to other features - e.g. women with heavy eye makeup, animals with long upward-pointing ears, or clusters of pink and purple flowers. SetGAN generates diverse output images that faithfully reproduce these features, whereas other baselines either copy the reference images or generate images which are not faithful to the shared features.

## 2.4 Set-based approaches in GANs

Ferrero et al. (2022) proposed an approach where the discriminator is allowed to make decisions based on a set of samples from either training data or the generator in order to increase stability and prevent mode collapse. While this work does also examine the idea of leveraging equivariances for generation, it focuses on improving the stability of unconditional generation rather than performing conditional set-based generation.

# 3 Methods

## 3.1 Background

### 3.1.1 Few-shot image generation

Few-shot image generation consists of a dataset $\mathcal{D}$ divided into a number of classes $\{\mathcal{C}_i\}$, which are each composed of some $n_{\mathcal{C}_i}$ images. These classes are partitioned into a disjoint training set $\mathcal{D}_{\text{train}}$ and test set $\mathcal{D}_{\text{test}}$. At inference time, a class $\mathcal{C} \in \mathcal{D}_{\text{test}}$ is sampled from the dataset. From this class, $k$ images are

sampled to become the *reference set* $\mathcal{C}_{\text{ref}}$, with the rest forming the holdout *evaluation set* $\mathcal{C}_{\text{eval}}$. The goal is to generate additional images $\mathcal{C}_{\text{gen}}|\mathcal{C}_{\text{ref}}$ such that the difference between the $\mathcal{C}_{\text{gen}}$ and $\mathcal{C}_{\text{eval}}$ is minimized, according to some sort of distance metric (e.g. the Frechet Inception Distance, see Section 4.4).

### 3.1.2 Conditional GANs

Generative Adversarial Networks (or GANs) typically follow an adversarial training paradigm in which two networks are trained jointly: a "discriminator" $D$ and a "generator" $G$. Given a dataset $\mathcal{D}$, these two networks train by playing a minimax game, often formulated approximately as follows:

$$\min_G \max_D \mathbb{E}_{x \sim \mathcal{D}} \log D(x) + \mathbb{E}_{z \sim p(z)} \log\left(1 - D(G(z))\right) \tag{1}$$

When applying a GAN to a conditional few-shot generation regime, this approach must be modified. A common way to proceed is to condition the generations on a single image. This means that the generator is no longer a mapping solely from the latent prior onto the data distribution $\mathcal{D}$, but rather a *conditional* mapping $G(z|x)$. The discriminator can then be viewed as a form of similarity function $D(x, y)$ between two images $x$ and $y$. During training, a class $\mathcal{C} \sim \mathcal{D}_{train}$ is sampled, and from this class are drawn two images $x, y \sim \mathcal{C}$. A modified minimax game is then played, of the form:

$$\min_G \max_D \mathbb{E}_{\mathcal{C} \sim \mathcal{D}} \mathbb{E}_{x,y \sim \mathcal{C}} [\log D(x, y) + \mathbb{E}_{z \sim p(z)} \log\left(1 - D(G(z|x), y)\right)] \tag{2}$$

This is the training regime used by methods such as DAGAN (Antoniou et al., 2017), DeltaGAN (Hong et al., 2020a), and AGE (Ding et al., 2022). While this does provide a method to train a conditional GAN, it also has limitations. By conditioning on a single image only, it can be difficult for the model to generate images that are faithful to the reference class. Many of these methods assume that the latent factors of variation within the classes at inference time will follow the same distribution as those of the training classes, and thus seek to perform transformations on the given input image to follow these factors of variation (Ding et al., 2022; Hong et al., 2020a).

### 3.1.3 Set-based models

In an ideal case, the model should be able to incorporate information from all reference images in order to better understand the latent space and generate diverse, high quality samples in a generalizable way. In order to do this, the model must be conditioned on a *set* of input images, rather than a single image. As such, the model must obey the restriction of *permutation equivariance* - i.e. for all permutations $\pi$ of the reference images $R$, $G(\pi(R)) = G(R)$.

The problem of constructing neural networks conditioned on sets of inputs while obeying restrictions of permutation-invariance or -equivariance has been discussed in previous works such as Zaheer et al. (2017), Lee et al. (2019) and Selby et al. (2022). As discussed in Lee et al. (2019), the simplest and most common model architecture which naturally conforms to these constraints is the transformer (Vaswani et al., 2017). The building block of the transformer is the so-called "attention mechanism", which takes the form

$$\text{MHA}(X, Y) = \sigma\left((XW_Q)(YW_K)^T\right) YW_V W_O \tag{3}$$

This structure is naturally permutation-equivariant with respect to the queries $X$ and permutation-invariant with respect to the keys Y, since for any permutation $\pi$, $\text{MHA}(\pi(X), Y) = \pi(\text{MHA}(X, Y))$ and $\text{MHA}(X, \pi(Y)) = \text{MHA}(X, Y)$. The "transformer decoder" architecture proposed by Vaswani et al. (2017) retains these properties, and constitutes a mapping $f : \mathbb{R}^{n \times d} \times \mathbb{R}^{m \times d} \to \mathbb{R}^{n \times d}$.

### 3.2 Set-GAN

Instead of conditioning on a single image only, SetGAN conditions its generations on a set of images from the same class, and seeks to generate a set of output images similar to these input images.

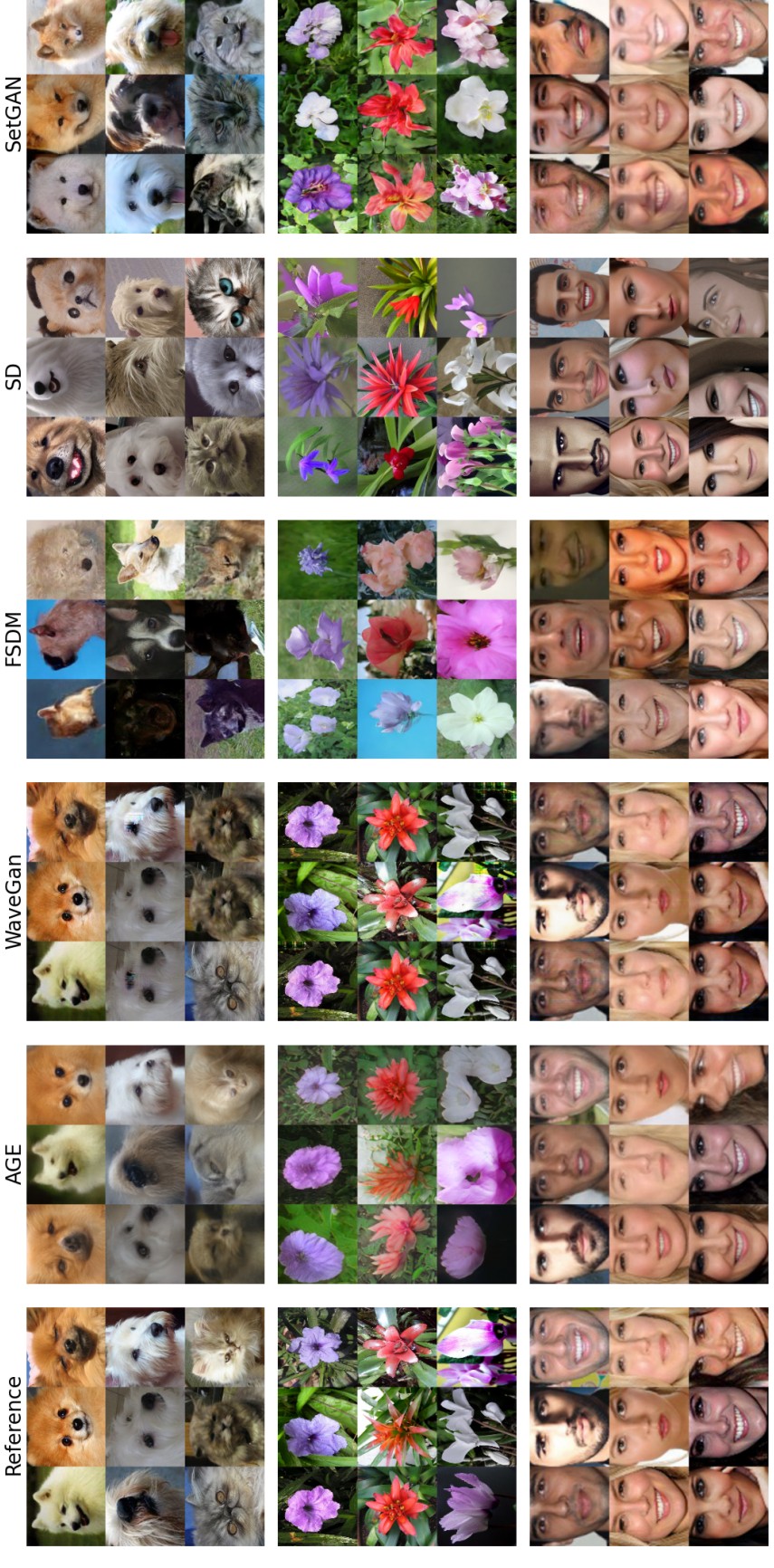

Figure 4: Generations from AGE, FSDM, WaveGAN and SetGAN conditioned on 3 reference images from unseen classes of each of the Animal Faces, Flowers and VGGFace datasets.

Formally, we again consider a setting where there is a dataset consisting of a number of classes $\{\mathcal{C}_i\}$, which are each composed of some $n_{\mathcal{C}_i}$ images. During training, a class $\mathcal{C} \sim \mathcal{D}_{train}$ is sampled, and from this class are drawn two (disjoint) sets of images: a *reference set* $R \in \mathcal{C}^n$, and a *candidate set* $C \in \mathcal{C}^m$. The generator $G$ produces $m$ generated images $G(R; m)$ conditioned on these reference images, which are then compared to the candidate set $C$ by a discriminator $D$, which plays the following minimax game with the generator:

$$\min_G \max_D \mathbb{E}_{\mathcal{C} \sim \mathcal{D}_{train}} \mathbb{E}_{R \sim \mathcal{C}^n, C \sim \mathcal{C}^m} \left[ \log D(R, C) \right] + \log\left(1 - D(R, G(R))\right) \tag{4}$$

### 3.3 Architecture

#### 3.3.1 Generator

The generator model follows an encoder-decoder structure similar to that of a U-Net (Ronneberger et al., 2015). We take StyleGAN2's generator to be our base decoder architecture, which maps a series of $k = 18$[1] 512-dimensional *style vectors* to a single output image, with each style vector controlling the convolutions at a particular stage of the decoding. It has become common to refer to the extended latent space formed by the concatenation of these $k$ vectors as $\mathcal{W}+$. Similar to Ding et al. (2022), we take the pixel2style2pixel (pSp) encoder proposed by Richardson et al. (2021) to be our encoder model, which maps a single input image into the space $\mathcal{W}+$ (although this could also be done with other similar encoders such as E4E (Tov et al., 2021) or ReStyle (Alaluf et al., 2021)). We augment this encoder-decoder model with a series of attention-based conditioning networks, consisting of a stack of 2 transformer decoder blocks for each of the $k$ style vectors, each surrounded by a skip connection.

Given a set of $n$ reference images, we encode each image $R_i$ into a latent code: $C_i = \text{pSp}(R_i) = \{c_i^0, ..., c_i^k\}$, with the notation $c_i^\ell$ for style $\ell$ of the encoding of image $i$, and $C^\ell = \{c_i^\ell\}$. To generate a set of $m$ candidate images, we then sample $m$ noise vectors $Z = z_{1,...,m} \sim N(0, 1)$. These are then passed through the decoder's mapping network to generate the base style vectors $W = \{f(z_j)\}$, in the same fashion as StyleGAN. Now, the model takes the base style vectors $W$ and transforms them by attending to the features of the reference encodings $C$. At each layer $\ell$, the model computes the corresponding *conditional style vector*:

$$\omega^\ell = g^\ell(W, T^\ell(W, C^\ell)) \tag{5}$$

where $T^\ell$ is the transformer block associated with the $\ell$-th style vector, and $g^\ell$ is a linear layer applied to the concatenation of the base style vector with the output of the attention blocks. These $k$ conditional style vectors then form our conditional encoding $\omega \in \mathcal{W}+$, which becomes the input to the StyleGAN2 decoder.

#### 3.3.2 Discriminator

The discriminator now takes the form $D : \mathbb{R}^{n \times d} \times \mathbb{R}^{m \times d} \to \mathbb{R}$, mapping an input reference set $R$ and candidate set $C$ to a single scalar output. To do this, we must use an architecture that can take as input multiple permutation-invariant sets. For this, we use the Multi-Set Transformer network proposed by (Selby et al., 2022). The input images are first passed through a convolutional encoder to encode each image within the two input sets as fixed-sized vectors, then passed through the multi-set transformer network. Finally, the two pooled output vectors are concatenated and fed to a linear output head. Skip connections are used to connect the outputs of the convolutional encoder to the latent vectors just before the pooling layer of the multi-set transformer. We use the convolutional architecture of the StyleGAN2 discriminator as the architecture for our discriminator encoder.

---

[1]Note that the default 18 style vectors correspond to a generation size of 1024x1024 px. Our experiments use a generation size of 256x256, and thus in practice use a truncated $\mathcal{W}+$ space of 14 vectors.

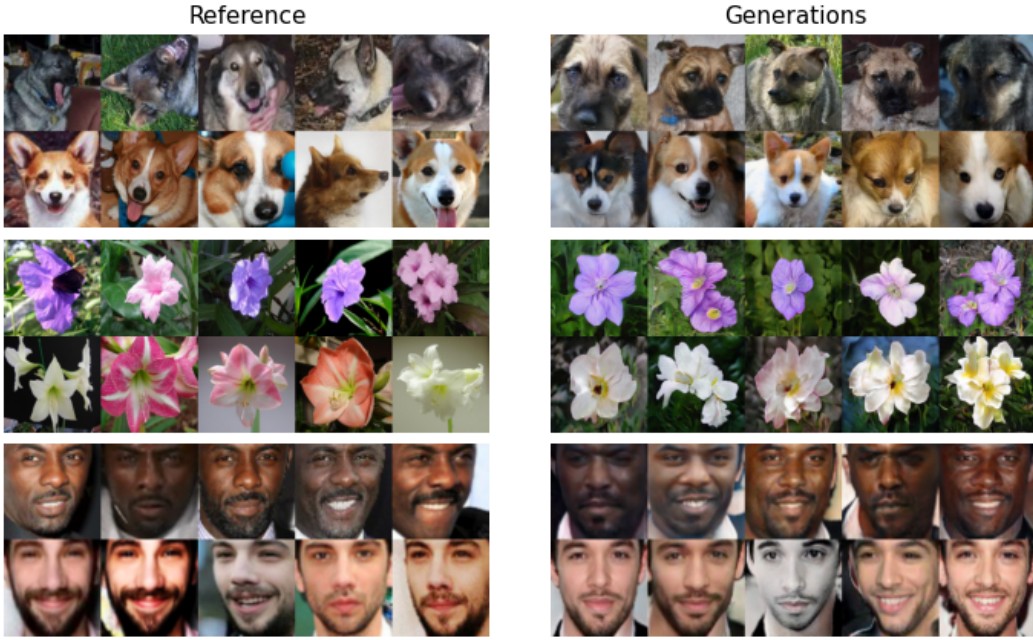

Figure 5: Additional generations from SetGAN using reference sets of 5 images.

## 4 Experiments

### 4.1 Setup

We first pretrain a StyleGAN2 model (Karras et al., 2020) on the given dataset at 256x256 resolution. Then, we train a pSp (Richardson et al., 2021) encoder to perform GAN inversion on the pretrained StyleGAN2 model to act as our encoder. These pretrained models are used to instantiate the encoder and decoder for our generator, and are then frozen. The discriminator from the StyleGAN2 model is also used to initialize the encoder for our multi-set discriminator model. These models are then trained following Eq. 4 until convergence. We use the base training scheme of StyleGAN2 (Karras et al., 2020) to train SetGAN, using a non-saturating loss with R1 gradient penalty ($\lambda = 10$) and path length regularization. Reference and candidate sizes are sampled uniformly from size 7-10 and 4-6 respectively, so that the model does not learn to assume a specific input size. Models are trained on NVIDIA A40 GPUs with the ADAM optimizer, with a batch size of 2 and learning rate 1e-3.

For inference, we follow similar models such as StyleGAN2 and apply latent space truncation, shifting the latent style vectors towards well-explored areas near the mean by a constant factor. Details of how this is applied are included in the supplementary material.

### 4.2 Datasets

In keeping with prior works (Hong et al., 2020a; Ding et al., 2022; Yang et al., 2022; Gu et al., 2021), we choose to report results on the Animal Faces (Liu et al., 2019a), Flowers (Nilsback & Zisserman, 2008) and VGGFace (Cao et al., 2018) datasets. We use the same train and evaluation splits proposed in Hong et al. (2020a) on Animal Faces and Flowers. For VGGFace, we restrict the evaluation set to the final 53 classes due to the computational requirements of inference for the FSDM baseline.

### 4.3 Baselines

Due to significant inconsistencies with existing results and methodologies (see appendix for details), we chose a selection of the highest performing models from the literature as baselines and computed metrics for each

| | MIFID$_{\text{Inc}}$ | | | LPIPS | | | F1 | | |
|---|---|---|---|---|---|---|---|---|---|
| | 1 | 3 | 10 | 1 | 3 | 10 | 1 | 3 | 10 |
| Animal Faces | | | | | | | | | |
| AGE | 71.35 | 62.23 | 56.55 | 0.4027 | 0.5095 | 0.5504 | 0.0901 | 0.2207 | 0.3791 |
| WaveGAN | 2327.29 | 1057.39 | 529.08 | 0.0000 | 0.4211 | 0.5556 | 0.0000 | 0.0004 | 0.0144 |
| FSDM | 75.68 | 73.93 | 77.37 | 0.6039 | 0.6076 | 0.6086 | 0.4425 | 0.4446 | 0.4318 |
| SD | *60.75* | 54.29 | 53.32 | 0.5703 | 0.5982 | 0.6081 | 0.3379 | 0.4359 | 0.4858 |
| SetGAN | 61.51 | *52.34* | *47.18* | *0.6144* | *0.6154* | *0.6181* | *0.4980* | *0.5333* | *0.5297* |
| Flowers | | | | | | | | | |
| AGE | 81.87 | 70.15 | 65.48 | 0.3790 | 0.5528 | 0.6078 | 0.0034 | 0.0034 | 0.0181 |
| WaveGAN | 2653.56 | 1305.31 | 699.96 | 0.0000 | 0.4844 | 0.6345 | 0.0000 | 0.0000 | 0.0005 |
| FSDM | 69.25 | 62.35 | 61.47 | 0.6809 | 0.6985 | 0.7042 | 0.1781 | 0.1789 | 0.1760 |
| SD | *54.56* | *50.98* | *52.66* | *0.7348* | *0.7546* | *0.7584* | *0.2405* | *0.2863* | *0.3163* |
| SetGAN | 62.44 | 59.84 | 59.31 | 0.6166 | 0.6240 | 0.6281 | 0.0217 | 0.0279 | 0.0275 |
| VGGFace | | | | | | | | | |
| AGE | 22.12 | 18.39 | 16.76 | 0.2604 | 0.3693 | 0.4063 | 0.0099 | 0.0346 | 0.1103 |
| WaveGAN | 852.7 | 36.97 | 23.12 | 0.0000 | 0.3246 | 0.4301 | 0.0000 | 0.0000 | 0.0014 |
| FSDM | 10.51 | 11.26 | 12.48 | 0.4509 | 0.4477 | 0.4471 | 0.3631 | 0.3332 | 0.3082 |
| SD | 51.64 | 52.63 | 52.75 | *0.5436* | *0.5568* | *0.5616* | 0.0625 | 0.0637 | 0.0675 |
| SetGAN | *9.60* | *7.93* | *7.83* | 0.4633 | 0.4614 | 0.4712 | *0.5468* | *0.5602* | *0.5723* |

Table 1: Scores for conditional generation on the Animal Faces, Flowers and VGGFace datasets for each of the five baselines, conditioned on reference sets of size 1, 3 and 10. Results were averaged over three different random partitions of the test set into $D_{\text{eval}}$ and $D_{\text{ref}}$. Lower scores are better for MIFID, higher is better for LPIPS and F1. The best score in each category is bolded. Scores that exceed all others by at least one standard deviation are italicized. Full results with standard deviations are reported in Tables 4, 6, 9. Precision and recall used to compute the F1 scores are reported in Tables 7, 8.

model ourselves by running the provided models under identical settings to ensure a fair comparison. Code and checkpoints provided by the authors were used wherever possible. We selected the AGE (Ding et al., 2022) and WaveGAN (Yang et al., 2022) models as representative of the highest-performing GAN-based approaches in the literature, as well as the diffusion-based approach FSDM (Giannone et al., 2022). To compare against a zero-shot diffusion model, we chose the Stable Diffusion Image Variations baseline, using the pretrained models provided[2].

## 4.4 Evaluation procedure and metrics

During evaluation, each test class $\mathcal{C} \in \mathcal{D}_{\text{test}}$ is partitioned into a reference set $\mathcal{C}_{\text{ref}}$ of size $n_{\text{ref}}$ and evaluation set $\mathcal{C}_{\text{eval}}$ of size $n_{\text{eval}}$. For each such class, the model is used to generate $n_{\text{gen}}$ new images, conditioned on images from $\mathcal{C}_{\text{ref}}$, to form $\mathcal{C}_{\text{gen}}$. For some metrics (such as FID or MIFID), these images are then aggregated into a single $D_{\text{eval}}$ and $D_{\text{gen}}$. These image sets are then used to evaluate the generations using a variety of metrics. For our experiments, $n_{\text{eval}} = n_{\text{gen}} = 128$, and $n_{\text{ref}}$ varied by experiment (see Section 5 for further details). If the number of images in a given evaluation set was lower than 128, all images were used. Each of these evaluations was performed three times with different randomly chosen partitions for each class.

The most common metrics used to evaluate the quality of models trained to perform few-shot image generation are the Frechet Inception Distance (FID) (Heusel et al., 2018), and Learned Perceptual Image Patch Similarity (LPIPS) (Zhang et al., 2018). FID measures the statistical similarity between distributions of embedded vectors corresponding to the evaluation set and generated sets respectively, and is often used as

---

[2]see https://huggingface.co/lambdalabs/sd-image-variations-diffusers

a measure of generation quality/fidelity. LPIPS is a metric used to measure perceptual similarity between pairs of images via the distance between their encodings under the pretrained VGG network. This is used as a metric for the diversity of generated images by computing the average pairwise distance between pairs of generated images within each class.

### 4.4.1   Limitations of existing metrics

While the aforementioned FID and LPIPS scores are the most widely-used metrics among existing literature, these metrics have significant flaws - particularly FID. Existing works such as Rangwani et al. (2023) and Kynkäänniemi et al. (2023) have already identified flaws in the FID metric related to its bias towards particular features specific to the ImageNet classes it was trained on, leading to arbitrary manipulation of scores via imperceptible changes in generated images. Rangwani et al. (2023) also demonstrate that traditional FID scores sometimes strongly emphasize fidelity over diversity in few-shot generation, and propose $\mathrm{FID_{CLIP}}$ in order to address this issue - a modification to the FID method using the large multi-modal CLIP model (Radford et al., 2021) in place of the Inception backbone.

While $\mathrm{FID_{CLIP}}$ is an improvement over traditional Inception-based FID scores in some respects, it does not wholly solve this problem. In our experiments, we found that models that generate identical or nearly identical copies of the reference images achieved extremely low FID scores. To test this, we measured the FID scores between the evaluation sets and generated sets constructed solely by copying N random images sampled with replacement from the reference set (denoted as the "Copy" baseline). We also tried the same experiment if the copied images were subjected to a small, imperceptible level of Gaussian noise (denoted as the "Noisy" baseline). We compare these scores to the best scores among all trained models[3], as well as a theoretical maximum score given by comparing two partitions of the test set. As shown in Table 2, this baseline of simply copying the reference images achieves FID scores close to the theoretical maximum, and matches or exceeds the score of the best trained baseline in almost every case. As a result, we conclude that *traditional FID scores are not a reliable metric for measuring the performance of few-shot generation.*

| | FID | $\mathrm{FID_{CLIP}}$ | MiFID | $\mathrm{MiFID_{CLIP}}$ |
|---|---|---|---|---|
| | Animal Faces | | | |
| Best Model | 46.20 | 5.02 | 46.20 | 5.02 |
| Noisy | 24.05 | 6.94 | 109.66 | 14.44 |
| Copy | 20.44 | 1.59 | 17714.97 | 1335.72 |
| True | 13.56 | 1.05 | 13.56 | 1.05 |
| | Flowers | | | |
| Best Model | 57.12 | 9.30 | 57.75 | 9.30 |
| Noisy | 37.06 | 4.21 | 394.61 | 22.37 |
| Copy | 36.98 | 2.56 | 23408.14 | 1737.27 |
| True | 30.18 | 1.69 | 30.18 | 1.69 |
| | VGGFace | | | |
| Best Model | 8.87 | 2.95 | 8.87 | 2.95 |
| Noisy | 47.96 | 12.20 | 56.51 | 12.65 |
| Copy | 9.54 | 0.77 | 4849.92 | 438.17 |
| True | 7.17 | 0.58 | 7.15 | 0.58 |

Table 2: Scores for synthetic baselines using a variety of performance metrics. Methods that simply copy the reference set ("Noisy" and "Copy") are disproportionately favored by many scoring methods, outperforming most trained models and even approaching the score for the true test set. MIFID scores are discussed in section 4.4.2.

### 4.4.2   Alternative metrics

These issues have also been identified in many other previous works, in the context of training set memorization. Works such as Gulrajani et al. (2020), Bai et al. (2021) and Jiralerspong et al. (2023) have discussed the tendency for traditional GAN evaluation metrics such as FID to overvalue fidelity and fail to penalize training set memorization. While these tools were proposed to measure generalization beyond the training set, they can also be equivalently applied here in order to measure generalization beyond the reference set. These works propose a wide range of different possible evaluation metrics as solutions to this problem, but most have key limitations that prevent them from being effective in this case. We choose to focus on MiFID

---

[3]We exclude WaveGAN from this, given WaveGAN's propensity to also generate nearly-identical copies of the reference images.

(Bai et al., 2021), but a more detailed discussion of the other approaches and their unsuitability for our purposes is included in the supplementary material.

**MIFID**

MiFID uses the standard Frechet Inception Distance, scaled by a multiplicative penalty calculated from the similarities between the generations and the reference images:

$$\text{MiFID}(S_g, S_t) = m_\tau(S_g, S_t) \cdot \text{FID}(S_g, S_t) \tag{6}$$

wherein $S_g$ is the generated set, $S_t$ is the training set (or reference set, in the case of conditional generation), FID is the standard Frechet Inception Distance, and $m_\tau$ is the penalty factor. Specifically, $m_\tau$ is defined by:

$$s(S_g, S_t) = \frac{1}{|S_g|} \sum_{x_g \in S_g} \min_{x_t \in S_t} 1 - \frac{|\langle x_g, x_t \rangle|}{|x_g| \cdot |x_t|} \tag{7}$$

$$m_\tau = \begin{cases} \frac{1}{s(S_g, S_t) + \epsilon} & s(S_g, S_t) < \tau \\ 1 & \text{else} \end{cases} \tag{8}$$

This metric penalizes models that simply reproduce reference images by adding a multiplicative penalty based on the average cosine similarity between the generated images and the nearest reference image. As shown in Table 2, this metric successfully penalizes models that simply copy the inputs, while keeping the original FID scores otherwise intact.

We adopt this metric as a drop-in replacement for FID, with the threshold $\tau$ determined by the average scale of the test set $S_{\text{test}}$. For each dataset, we divide the test set into two partitions of equal size, $S_1$ and $S_2$, then calculate the base score value $\tau_0 = s(S_1, S_2)$ using Equation 7. To ensure that models which produce results on a similar scale of variation to the test set are not unfairly penalized, we penalize only models whose scores are at least one standard deviation lower than the mean similarity scale (i.e. $\tau = \tau_0 - \sigma$, where $\sigma$ is the standard deviation of the summand in Eq. 7).

### 4.4.3 Precision, Recall and F1 Score

In addition to metrics based on FID or LPIPS, we may also consider other measures of generation quality such as the precision and recall measures suggested by Kynkäänniemi et al. (2019). The authors propose to extend the commonly-used notions of precision and recall for classification into the domain of image generation by using nearest-neighbours approximations to measure the overlapping supports between the distributions $D_{\text{eval}}$ and $D_{\text{gen}}$. We include the F1 score computed by harmonic average of precision and recall as an additional metric to assess the overall alignment of the distributions.

## 5 Results

### 5.1 Quantitative results

Results for all baselines are shown in Table 1 using MIFID, LPIPS and F1 score (Results for $\text{MIFID}_{\text{CLIP}}$ as well as precision and recall individually are included in the appendices). Results are shown for reference sizes of 1, 3 and 10, across each of the 3 datasets. As shown in the table, SetGAN achieves superior performance to all other baselines on almost all settings of the Animal Faces and VGGFace datasets in terms of generation quality and fidelity (as measured by MIFID and F1 score). Its results are also highly diverse, achieving LPIPS scores greater than all other baselines on the AnimalFaces dataset, and greater than all save the Stable Diffusion baseline on the VGGFace dataset.

While SetGAN's performance does suffer on the Flowers dataset compared to the Stable Diffusion baseline, it still outperforms all other baselines in terms of MIFID scores, and all save FSDM in terms of LPIPS and

F1. It is important to note while the FSDM approach performs well numerically on this dataset, it generates images with very poor fidelity to the reference class. This is likely a large part of the reason for its increased diversity (see Section 5.2), resulting in high LPIPS scores but notably worse FID scores for the FSDM model across most datasets. In addition, the success of the Stable Diffusion baseline on this dataset relative to the other baselines is likely due to the dataset's small size. All other baselines on this dataset were limited by the very small amount of training data. Stable Diffusion, on the other hand, was pretrained on very large datasets of general image data, and thus not affected by this limitation. This did have other consequences, however - Stable Diffusion's generations often had a cartoon-like quality (see Section 5.2), which did not match well with the domain of the VGGFace dataset, resulting in very poor performance.

Notably, WaveGAN performs markedly worse than the other models under the MiFID score - largely due to it being heavily penalized for its tendency to produce nearly-indistinguishable copies of the reference images (see 5.2). While other models such as AGE often produce images very similar to the reference images, they were not copies, and as such did not fall under the threshold to be penalized.

## 5.2 Qualitative results

### 5.2.1 Test images with similar factors of variation to the training classes

Figure 4 shows images generated by the five models conditioned on reference images from the test classes of each of the three datasets considered. For these experiments, all reference images were drawn from the same unseen test class - measuring the models' effectiveness at generalization along similar factors of variation to the training classes. As shown in the figure, SetGAN generates diverse, high quality images, and avoids many of the struggles that other models demonstrate. Models such as AGE and WaveGAN often simply copy one of the input images, or generate small, subtle variations on it. This causes their generations to be limited in diversity, particularly when conditioned on only a small number of images. WaveGAN in particular very frequently copies the reference image almost exactly, differing from it only in imperceptible high-frequency perturbations. FSDM does succeed at generating diverse images, but often struggles to closely match the input class. This is particularly notable in the results from the flowers dataset, where its generations were often starkly different from the reference class. The Stable Diffusion baseline also generates high-quality images that are relatively true to the reference classes, but sometimes struggles in other ways. Its generations often have a cartoonish quality - particularly those on the VGGFace dataset - and many are oddly skewed, or cut off in places.

### 5.2.2 Test images with different factors of variation from the training classes

In addition to evaluating the models' generations given images from the same unseen class, we can also examine how the models perform given images from *different* classes. Rather than grouping reference images by their original class in the dataset (i.e. the type of animal, type of flower, or individual person), we selected three groups of images wherein each image was taken from a different class, but all images shared common traits. In the first example, the images consisted of animals of many different types or breeds, with the shared trait being long upward-pointing ears. The second example contained images of flowers from many different types, all of which contained clusters of multiple pink or purple flowers. The third contained images of many different women who were all wearing bold, dark eyeshadow. The resulting generations are shown in Figure 3.

In all cases, SetGAN accurately reproduced the target features while generating a diverse range of output images. Other baselines which conditioned on a single image (i.e. AGE and WaveGAN) each struggled with this - again generating output images either identical to the inputs or very similar with subtle variations. These subtle variations would sometimes lead to deviations from the target features, as the models did not have multiple images to compare to in order to identify which features were shared. For example, the generations from AGE led to some images with short ears, single flowers, or less distinctive makeup. The FSDM baseline *was* also capable of incorporating features from multiple images, but the results were often of lower quality and were less faithful to the target features than those of SetGAN. The Stable Diffusion baseline also performed well on this task, and generally produced results sharing many of reference features.

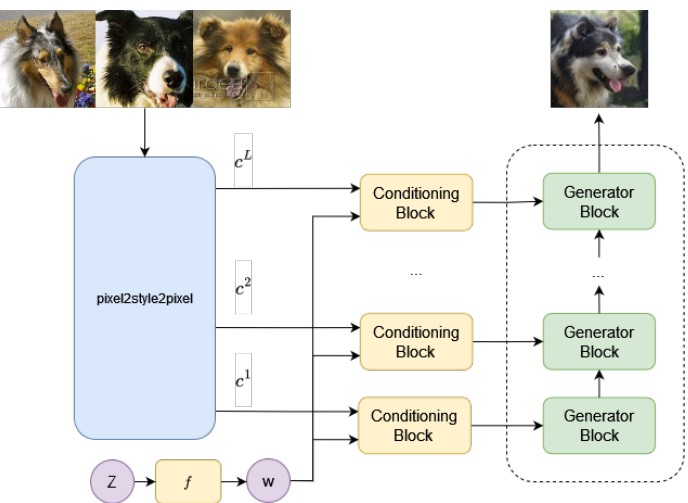

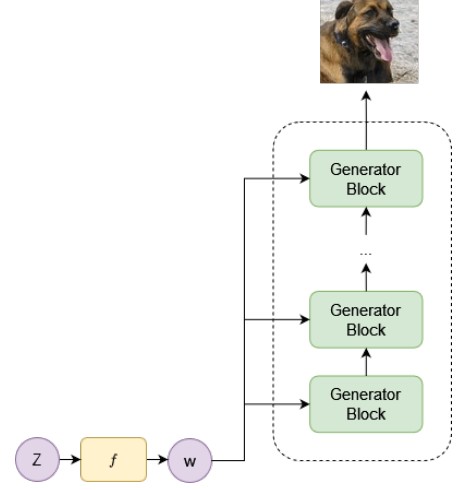

(a) The full generation process for a given style $w$ and set of reference images $R$ using SetGAN.

(b) The "template image" generated by the same $w$ with no conditioning layers.

Figure 6: Diagrams of an example generation process from SetGAN.

As was discussed in Section 5.2.1, however, its generations often had other issues, such as strange degrees of rotation, or images shifted strongly to one side.

# 6 Analysis

In order to visualize how SetGAN constructs an output image from a given set of inputs, consider the example generation shown in Fig. 6a. As explained in section 3.3.1, the generation process begins by encoding each of these reference images into a latent representation $C_i$ using the pSp encoder. The model will then generate a series of $m$ Gaussian noise vectors (one per output image) and pass these through the pretrained StyleGAN2 mapping network to obtain base latent codes $W \in \mathcal{W}$. If these latent codes were fed directly to the generator, they would result in samples from the pretrained StyleGAN2 model, without any conditioning. We can consider these unconditional generations to be "template" images, which will then be transformed and modified to become the final output (see Fig. 6b).

In order to incorporate the information from the reference images, a series of attention-based conditioning layers will then combine the base latent codes $W$ with the reference encodings $c_i^\ell$ at each layer of the network to produce a series of *conditional* style vectors $\omega$. This will have the effect of progressively shifting the template image towards the reference images as it progresses through the network.

## 6.1 Effect of the conditioning network by reference image

Figure 7 shows the relative weight given by the attention blocks in the conditioning network to each of the reference images from the generation process in Figure 6a. In order to visualize how the different weight for each image affects the generation outputs, let us consider the effects of the conditioning network on just a single style. With the conditioning network active on only the first style vector, Figure 8 shows examples of the output with varying degrees of weight given to each reference image - including examples with the true weights taken from the heatmap in Figure 7, as well as 100% weight given to each reference image in turn.

The effect of the varying attention weights at this layer on the final image can be clearly seen from these examples. Features such as the ear shape, ear orientation,

Figure 7: Heatmap of attention weights by layer for Fig. 6a

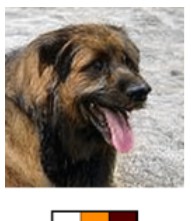 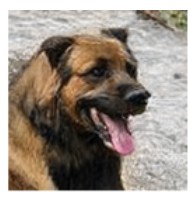 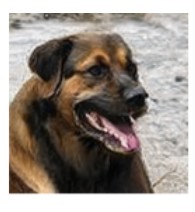 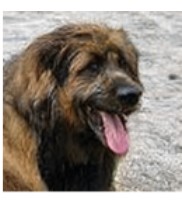

Figure 8: Sample generations using the reference images in Figure 6 with only the first conditioning layer active. Heatmaps underneath each image indicate the attention weights given to each reference image.

fur texture and tongue/mouth position change significantly in accordance with the reference image being most closely attended to at this layer. The effect can be clearly seen on those same features in the final output image. The ears take on a slightly rounded shape, the fur texture becomes shaggy and long, and the open mouth takes on a slight upward lilt that looks almost like a smile - all features strongly similar to the third reference image. This matches the values shown in Figure 7, where the weights are indeed highly concentrated around that same image.

## 6.2 Effect of the conditioning network by layer

These previous examples highlighted the effects of the attention layers in attending to and incorporating features from the reference images - but only using a single layer. To see the cumulative effects of these conditioning layers throughout the generation process, we apply the generation process with a variable number of conditioning layers active. As before, inactive layers use only the base style vector as input. The results of this experiment are shown in Figure 9. Initially, no conditioning layers are active, and the generator produces the template image mentioned previously. As more layers are introduced, additional features from the reference image are used to adjust this template image further and further towards the images in the reference set.

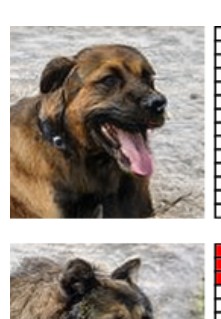

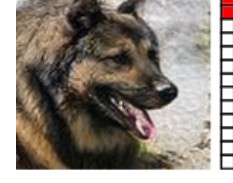

Interestingly, the features affected by the introduction of the conditioning layers vary strongly by the position of the layer in the network. Enabling the conditioning layers in the early layers affects coarse features such as fur texture, stripes/patches, head facing and ear position. In contrast, the middle conditioning layers affect the background, fur color, and finer adjustments to face structure/expression. Finally, the last layers in the network affect subtler qualities like color saturation and fine textural details.

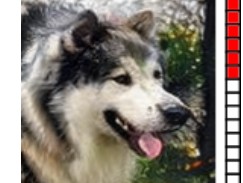

This matches closely with the common observation that the layers of the Style-GAN2 network affect the properties of the output image based on their location in the network, with earlier layers affecting coarser features of the image and later layers affecting the finer details. As our decoder is directly based on the StyleGAN2 decoder, it is unsurprising to observe the same property here.

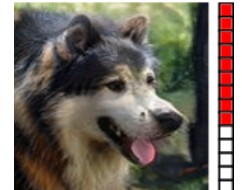

## 6.3 Effect of the base style vector

One interesting consequence of the many residual or skip connections through SetGAN's architecture is the predominant role played by the base style vector in the generation. As discussed previously, this base style vector represents a sort of "template image", that will then be modified by each of the conditioning layers in turn to attend to the features of the reference images. Despite the significant effects of these layers shown in the previous sections, the initial template image retains a strong effect on the final generation. Figure 10 shows

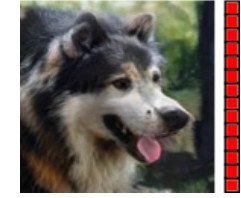

Figure 9: Generations from Fig. 6 with some attention layers inactive. Red boxes indicate active layers.

a series of generations using the same base style vector as in Fig. 6, but different reference images. Notice how all of these images retain similar features in terms of their orientation, head position and overall expression.

To understand the reason for this, consider Equation 6.3, which shows how the conditional encodings are incorporated into the styles:

$$\omega^\ell = g^\ell(W, T^\ell(W, C))$$

In this equation, $g^\ell$ represents a learned transform applied to the concatenation of the base style vector with the conditional style computed by the appropriate attention block. At the beginning of training, $g^\ell$ is initialized to act as an identity map on the base style, making this essentially a residual connection. As such, the computed conditional encoding will act as an offset *relative* to the base style - anchoring the output generation strongly to the template image.

# 7 Conclusion and Future Work

The task-specific experiments shown in this paper demonstrate that SetGAN can effectively replicate and even surpass the ability of other GAN-based approaches to learn the factors of variation within different classes in a dataset and generalize them to new classes at inference time. Its performance on these tasks can even compete with those of powerful pretrained diffusion models such as Stable Diffusion, despite being trained on only small, task-specific datasets. In addition, SetGAN shows potential to generalize beyond the structure of the training classes and flexibly perform generation conditioned on reference images sharing features across a wide array of different axes of similarity. We hope that in the future, this may be extended to more truly general, zero-shot forms of image generation on larger and more diverse datasets. Other approaches such as Giannone et al. (2022) have shown results on datasets such as CIFAR-100 and Mini-ImageNet, but these datasets are low-resolution and contain a very limited number of classes, which limits the model's ability to generalize to truly diverse and varied unseen classes at inference time. While Giannone et al. (2022) do report some succesful results at few-shot generation with these datasets, they often struggle to adapt to unseen classes at inference time as a result of this, and end up producing samples from unrelated training classes. Instead, our focus is on scaling our approach to truly diverse and large-scale high resolution datasets such as ImageNet. This may provide a path to achieving truly zero shot set-based image-to-image generation, and will be the focus of future work.

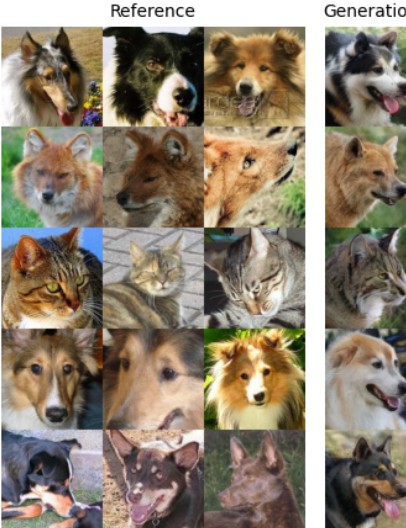

Figure 10: Generations from different reference batches using the same base style.

**Broader Impact Statement**

Image generation models can often be ethically fraught. Conditional text-to-image generative models have been the focus of significant uproar, both from those in the AI community and outside of it. The use of large-scale online image datasets has incited controversy due to intellectual property concerns and the alleged role these models play in disenfranchising artists. There are also ongoing concerns about the risk of generative models enabling the spread of disinformation, fake news and propaganda due to the difficulty in distinguishing AI-generated content from that which is human-generated. Image generation models can also be used to create so-called "deepfakes", and may be used to generate misleading or obscene content featuring the likenesses of real individuals.

All experiments performed in this paper are of limited scope and are unlikely to lead to major ethical challenges in the manner of their use. These experiments also do not leverage the large-scale online image

data that have elicited accusations of intellectual property theft. That being said, given that SetGAN has the potential to be scaled to a model with much broader and more general scope, it will become very important to be mindful of these concerns as we move forward.

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

# A  Appendix

## A.1  Architecture and training details

For our experiments, we used pretrained StyleGAN2 (Karras et al., 2020) and pSp (Richardson et al., 2021) models provided by Ding et al. (2022) [4]. We take the layers of these models corresponding to resolutions of 256x256 and lower (i.e., the first 14 layers) to use for our initialization. We use a standard StyleGAN2 generator as our decoder, with 14 layers and a latent dimension of 512. The mapping network uses 8 feed-forward layers, with a learning rate multiplier of 0.01. We use a standard pixel2style2pixel (pSp) encoder architecture, truncated to 14 style vectors. Conditioning networks in the generator used stacks of 2 transformer decoder blocks (see Fig. 1 in the main paper), with 16 attention heads and latent size 512. The discriminator used the standard StyleGAN2 discriminator architecture as its encoder, with the last layer removed. The multi-set transformer model in the discriminator used a stack of 4 multi-set attention blocks, with mean pooling, skip connections surrounding the multi-set transformer network, and a linear output projection. Training was performed using the StyleGAN2 training schema, with a nonsaturating loss, R1 gradient penalty ($\lambda = 10$), path length regularization ($\lambda = 2$) and style mixing ($p = 0.9$). We use lazy regularization, with the R1 penalty applied every 16 steps, and path length regularization applied every 4 steps. We use an exponential moving average of the generator weights, with $\gamma = 0.999$. Training, validation and test splits for each dataset followed the standard splits in prior work, except as discussed previously (see Ding et al. (2022); Hong et al. (2020a); Gu et al. (2021); Hong et al. (2020b)).

Experiments were performed using NVIDIA A40 GPUs. Each model was trained for approximately 1 week using 2 GPUs.

## A.2  Latent space truncation

SetGAN uses latent space truncation for inference, in a similar manner to StyleGAN2. In order to improve the quality of the generated results, style vectors are shifted towards the mapping network's mean style vector $\bar{w}$ by a given factor $\lambda$. Unlike StyleGAN2, however, this truncation may be applied to SetGAN in two ways: either pre-conditioning or post-conditioning.

---

[4] https://github.com/unibester/AGE

Given a base style vector $w$, pre-conditioning truncation is applied in the same manner as it is for StyleGAN: the latent vector is transformed by the procedure:

$$w \rightarrow \bar{w} + \lambda_1(w - \bar{w}) \tag{9}$$

This ensures that the base style vector used to generate the output images remains in the well-explored region near the mean, and leads to generations of higher quality but slightly lower diversity.

In addition to this, however, truncation may also be applied post-conditioning, to shift the final conditional styles $w'$ towards the mean style vector as follows:

$$w'_j \rightarrow \bar{w} + \lambda_2(w'_j - \bar{w}) \tag{10}$$

This has a large effect on output quality, but at a much greater cost to output diversity.

Both $\lambda_1$ and $\lambda_2$ truncation provided significant benefits on the Flowers dataset, improving sample quality and MiFID score by considerable amounts. $\lambda_1$ truncation improved sample quality and MiFID score for the Animal Faces dataset, but came at a cost of sample diversity. Truncation provided substantially less benefit on the VGGFace dataset than the other two, but a small level was still found to be beneficial for sample quality. Results in this paper were obtained with $\lambda_1 = \lambda_2 = 0.8$ for the Flowers dataset, $\lambda_1 = 0.8$ for the AnimalFaces dataset, and $\lambda_1 = 0.9$ for the VGGFace dataset.

### A.3    Inconsistencies in Prior Results

Many existing few-shot image generation models contain significant inconsistencies in the methodologies used for evaluation. For example, the LPIPS metric can be evaluated using either AlexNet or VGG activations, which cannot be compared directly against each other. We found that previous works such as F2GAN and DeltaGAN used AlexNet activations to measure LPIPS score, while WaveGAN and LoFGAN used VGG activations. These previous works also generated results at a variety of different resolutions - and some were then rescaled before applying the metric, while others were not. AGE generated outputs at 256x256, while other works performed their generations at 128x128. WaveGAN, LoFGAN and AGE also rescaled images to 32x32 before computing LPIPS distances, while other works did not. Different works also used different code for compiling generated images and rescaling them to the target size of the VGG, AlexNet or Inception models used to obtain vector embeddings. As discussed in Parmar et al. (2022), the details of these steps can have a substantial impact on the final results, and inconsistent methodologies between papers can lead to significant discrepancies. In addition to these inconsistencies in methodology, we found that in many cases we were unable to reproduce the reported scores of existing works - despite using code and checkpoints provided by the authors, and consulting with the authors directly.

### A.4    Discussion of additional evaluation metrics

As discussed in section 4.4.2, there were many existing candidates in the literature for evaluation metrics that were sensitive to training/reference set memorization. The four most notable candidates were Conditional FID (Soloveitchik et al., 2022), neural network divergences (Gulrajani et al., 2020), MiFID (Bai et al., 2021) and Feature Likelihood Score Jiralerspong et al. (2023). MiFID was discussed in section 4.4.2. Of the remaining metrics, conditional FID (Soloveitchik et al., 2022) requires an FID calculation to be computed over the reference set, which does not work for cases with small reference sizes due to the instability of the FID calculation with small numbers of samples. Even reference sizes of 10 would only allow for 200-500 samples (depending on dataset) - far too few to perform the FID calculation. Neural network divergences (Gulrajani et al., 2020) are architecture-specific, and must be trained repeatedly for each inference setting. This makes them different to compare across different models and publications, as well as costly to evaluate.

### A.4.1    Feature Likelihood Score

Finally, the last important candidate to consider is Feature Likelihood Score (FLS). Feature Likelihood Score uses a method similar to Kernel Density Estimation to fit a Gaussian Mixture density to the generated samples. The covariances of the mixture components are chosen to maximize the likelihood of the reference

| | $\text{FID}_{\text{Inc}}$ | $\text{FID}_{\text{CLIP}}$ | $\text{MiFID}_{\text{Inc}}$ | $\text{MiFID}_{\text{CLIP}}$ | $\text{FLS}_{\text{Inc}}$ | $\text{FLS}_{\text{CLIP}}$ |
|---|---|---|---|---|---|---|
| | | | Animal Faces | | | |
| Best Model | 46.20 | 5.02 | 46.20 | 5.02 | 125.93 | 133.38 |
| Noisy | 24.05 | 6.94 | 109.66 | 14.44 | 126.72 | 143.02 |
| Copy | 20.44 | 1.59 | 17714.97 | 1335.72 | 229.81 | 168.31 |
| True | 13.55 | 1.05 | 13.56 | 1.05 | 114.93 | 127.70 |
| | | | Flowers | | | |
| Best Model | 57.12 | 9.29 | 57.75 | 9.29 | 142.59 | 144.41 |
| Noisy | 37.06 | 4.21 | 394.61 | 22.37 | 144.18 | 143.83 |
| Copy | 36.98 | 2.56 | 23408.14 | 1737.27 | 164.80 | 169.98 |
| True | 30.18 | 1.69 | 30.18 | 1.69 | 139.21 | 132.39 |
| | | | VGGFace | | | |
| Best Model | 8.87 | 2.95 | 8.87 | 2.95 | 134.90 | 129.20 |
| Noisy | 47.96 | 12.20 | 56.51 | 12.64 | 148.23 | 145.64 |
| Copy | 9.54 | 0.77 | 4849.92 | 438.17 | 170.63 | 176.04 |
| True | 7.15 | 0.58 | 7.15 | 0.58 | 134.58 | 119.17 |

Table 3: Scores for all metrics (including FLS) on synthetic baselines.

set, ensuring that the density will be highly concentrated if the samples are simply copied from the reference data. The score is then calculated by evaluating the likelihood of the test data under this density. This scoring method is an interesting candidate, but fails to sufficiently penalize copying - particularly in cases where imperceptible perturbations are applied to the copied image. As shown in Table 3, the FLS scores for the "noisy" synthetic baseline nearly match those of the best trained models across multiple datasets.

### A.5  Full Results

Tables 4, 5, 6, 7, 8, 9 show the mean and standard deviation for all rresults for each of the 6 metrics.

| | MIFID$_{\text{Inc}}$ | | |
|---|---|---|---|
| | 1 | 3 | 10 |
| Animal Faces | | | |
| AGE | 71.3506 ± 5.57 | 62.2324 ± 3.33 | 56.5487 ± 0.85 |
| WaveGAN | 2327.29 ± 215.18 | 1057.39 ± 73.21 | 529.08 ± 14.99 |
| FSDM | 75.68 ± 0.92 | 73.93 ± 1.06 | 77.37 ± 2.40 |
| SD | 60.75 ± 5.15 | 54.29 ± 2.47 | 53.3177 ± 2.07 |
| SetGAN | 61.5095 ± 2.5821 | 52.3416 ± 0.9705 | 47.1835 ± 0.6288 |
| Flowers | | | |
| AGE | 81.87 ± 5.63 | 70.15 ± 2.46 | 65.48 ± 1.53 |
| WaveGAN | 2653.56 ± 143.46 | 1305.31 ± 20.58 | 699.96 ± 28.59 |
| FSDM | 69.25 ± 2.37 | 62.35 ± 0.29 | 61.47 ± 0.78 |
| SD | 54.56 ± 4.45 | 50.98 ± 1.49 | 52.66 ± 2.42 |
| SetGAN | 62.44 ± 2.27 | 59.84 ± 0.34 | 59.31 ± 0.92 |
| VGGFace | | | |
| AGE | 22.12 ± 1.33 | 18.39 ± 0.18 | 16.76 ± 0.30 |
| WaveGAN | 852.70 ± 672.02 | 36.97 ± 0.80 | 23.12 ± 0.15 |
| FSDM | 10.51 ± 0.37 | 11.26 ± 0.25 | 12.48 ± 0.13 |
| SD | 51.63 ± 2.40 | 52.63 ± 1.32 | 52.75 ± 0.31 |
| SetGAN | 9.60 ± 0.36 | 7.93 ± 0.41 | 7.83 ± 0.03 |

Table 4: MIFID$_{\text{Inc}}$ results on all datasets.

| | MIFID$_{\text{CLIP}}$ | | |
|---|---|---|---|
| | 1 | 3 | 10 |
| Animal Faces | | | |
| AGE | 14.0896 ± 1.077 | 12.7704 ± 0.7982 | 11.7434 ± 0.2656 |
| WaveGAN | 603.4436 ± 46.598 | 242.805 ± 6.3307 | 136.0878 ± 2.2945 |
| FSDM | 8.7805 ± 0.397 | 8.5878 ± 0.3752 | 10.3751 ± 0.8405 |
| SD | 11.8816 ± 0.6214 | 10.293 ± 0.3349 | 9.7812 ± 0.2608 |
| SetGAN | 6.5616 ± 0.3751 | 5.8394 ± 0.2124 | 5.2808 ± 0.0814 |
| Flowers | | | |
| AGE | 16.8227 ± 0.8373 | 15.0334 ± 0.511 | 14.3053 ± 0.1992 |
| WaveGAN | 851.1427 ± 30.1413 | 373.6241 ± 21.4988 | 182.1069 ± 5.865 |
| FSDM | 10.6865 ± 0.2972 | 10.2571 ± 0.2653 | 10.1823 ± 0.0252 |
| SD | 10.2983 ± 0.8236 | 8.9765 ± 0.254 | 8.6868 ± 0.2274 |
| SetGAN | 10.6834 ± 0.9376 | 9.7909 ± 0.406 | 9.8773 ± 0.2136 |
| VGGFace | | | |
| AGE | 8.1984 ± 0.3228 | 6.5102 ± 0.2239 | 5.9442 ± 0.1536 |
| WaveGAN | 17.4993 ± 0.757 | 9.3999 ± 0.2776 | 6.6542 ± 0.1239 |
| FSDM | 3.277 ± 0.1328 | 3.4727 ± 0.0789 | 3.7625 ± 0.0667 |
| SD | 18.8204 ± 0.2751 | 17.6513 ± 0.1792 | 17.0962 ± 0.1232 |
| SetGAN | 4.158 ± 0.2303 | 3.1172 ± 0.1376 | 2.8229 ± 0.0591 |

Table 5: MIFID$_{\text{CLIP}}$ results on all datasets.

| | LPIPS | | |
|---|---|---|---|
| | 1 | 3 | 10 |
| | Animal Faces | | |
| AGE | 0.4027 ± 0.0026 | 0.5095 ± 0.005 | 0.5504 ± 0.0023 |
| WaveGAN | 0.0000 ± 0.0000 | 0.4211 ± 0.0037 | 0.5556 ± 0.0022 |
| FSDM | 0.6039 ± 0.0006 | 0.6076 ± 0.0011 | 0.6086 ± 0.0006 |
| SD | 0.5703 ± 0.003 | 0.5982 ± 0.0057 | 0.6081 ± 0.0029 |
| SetGAN | 0.6144 ± 0.0019 | 0.6154 ± 0.0007 | 0.6181 ± 0.0007 |
| | Flowers | | |
| AGE | 0.379 ± 0.0114 | 0.5528 ± 0.0044 | 0.6078 ± 0.0037 |
| WaveGAN | 0.0000 ± 0.0000 | 0.4844 ± 0.0044 | 0.6345 ± 0.0031 |
| FSDM | 0.6809 ± 0.0038 | 0.6985 ± 0.0026 | 0.7042 ± 0.001 |
| SD | 0.7348 ± 0.0038 | 0.7546 ± 0.0011 | 0.7584 ± 0.0012 |
| SetGAN | 0.6166 ± 0.0032 | 0.624 ± 0.0037 | 0.6281 ± 0.0019 |
| | VGGFace | | |
| AGE | 0.2604 ± 0.0005 | 0.3693 ± 0.0043 | 0.4063 ± 0.0028 |
| WaveGAN | 0.0000 ± 0.0000 | 0.3246 ± 0.0029 | 0.4301 ± 0.0036 |
| FSDM | 0.4509 ± 0.0034 | 0.4477 ± 0.0014 | 0.4471 ± 0.0006 |
| SD | 0.5436 ± 0.0049 | 0.5568 ± 0.0047 | 0.5616 ± 0.0022 |
| SetGAN | 0.4633 ± 0.0049 | 0.4614 ± 0.0038 | 0.4712 ± 0.0008 |

Table 6: LPIPS results on all datasets.

| | Precision | | |
|---|---|---|---|
| | 1 | 3 | 10 |
| | Animal Faces | | |
| AGE | 0.5363 ± 0.0659 | 0.5187 ± 0.0415 | 0.5332 ± 0.0177 |
| WaveGAN | 0.6358 ± 0.1169 | 0.684 ± 0.0502 | 0.6661 ± 0.0452 |
| FSDM | 0.5227 ± 0.0533 | 0.5323 ± 0.0503 | 0.5286 ± 0.0357 |
| SD | 0.5022 ± 0.0528 | 0.5204 ± 0.0600 | 0.5367 ± 0.0546 |
| SetGAN | 0.5394 ± 0.0413 | 0.6011 ± 0.0296 | 0.6393 ± 0.0168 |
| | Flowers | | |
| AGE | 0.4913 ± 0.0016 | 0.4776 ± 0.0221 | 0.4883 ± 0.0333 |
| WaveGAN | 0.6861 ± 0.1167 | 0.7201 ± 0.0919 | 0.6444 ± 0.043 |
| FSDM | 0.326 ± 0.0268 | 0.3427 ± 0.0041 | 0.3764 ± 0.0125 |
| SD | 0.3209 ± 0.0293 | 0.2969 ± 0.002 | 0.3164 ± 0.0046 |
| SetGAN | 0.4716 ± 0.0168 | 0.5571 ± 0.0448 | 0.5799 ± 0.0508 |
| | VGGFace | | |
| AGE | 0.7499 ± 0.0374 | 0.7393 ± 0.0215 | 0.7419 ± 0.0091 |
| WaveGAN | 0.4791 ± 0.0228 | 0.4498 ± 0.0349 | 0.4785 ± 0.0041 |
| FSDM | 0.7613 ± 0.0049 | 0.7738 ± 0.0048 | 0.7749 ± 0.0036 |
| SD | 0.0427 ± 0.0026 | 0.0429 ± 0.0027 | 0.0425 ± 0.0019 |
| SetGAN | 0.6075 ± 0.0108 | 0.6306 ± 0.0061 | 0.6268 ± 0.0066 |

Table 7: Precision results on all datasets.

| | Recall | | |
|---|---|---|---|
| | 1 | 3 | 10 |
| Animal Faces | | | |
| AGE | 0.0493 ± 0.0061 | 0.1411 ± 0.0244 | 0.2947 ± 0.0322 |
| WaveGAN | 0.0000 ± 0.0000 | 0.0002 ± 0.0003 | 0.0073 ± 0.0038 |
| FSDM | 0.3842 ± 0.0133 | 0.3837 ± 0.0466 | 0.3687 ± 0.0598 |
| SD | 0.2616 ± 0.0634 | 0.3771 ± 0.0094 | 0.4479 ± 0.0349 |
| SetGAN | 0.4625 ± 0.0339 | 0.4793 ± 0.0618 | 0.4522 ± 0.0052 |
| Flowers | | | |
| AGE | 0.0017 ± 0.0015 | 0.0017 ± 0.0011 | 0.0092 ± 0.0024 |
| WaveGAN | 0.0000 ± 0.0000 | 0.0000 ± 0.0000 | 0.0003 ± 0.0005 |
| FSDM | 0.1229 ± 0.0159 | 0.1214 ± 0.0161 | 0.115 ± 0.0061 |
| SD | 0.1939 ± 0.0175 | 0.2772 ± 0.0266 | 0.3163 ± 0.0087 |
| SetGAN | 0.0111 ± 0.0036 | 0.0143 ± 0.0054 | 0.0141 ± 0.0014 |
| VGGFace | | | |
| AGE | 0.005 ± 0.0022 | 0.0177 ± 0.0015 | 0.0596 ± 0.002 |
| WaveGAN | 0.0000 ± 0.0000 | 0.0000 ± 0.0000 | 0.0007 ± 0.0004 |
| FSDM | 0.2384 ± 0.0026 | 0.2123 ± 0.0081 | 0.1927 ± 0.0200 |
| SD | 0.1164 ± 0.0148 | 0.1238 ± 0.0136 | 0.1638 ± 0.0244 |
| SetGAN | 0.4972 ± 0.0100 | 0.5040 ± 0.0087 | 0.5265 ± 0.0088 |

Table 8: Recall results on all datasets.

| | F1 | | |
|---|---|---|---|
| | 1 | 3 | 10 |
| Animal Faces | | | |
| AGE | 0.0901 ± 0.0103 | 0.2207 ± 0.0293 | 0.3791 ± 0.0291 |
| WaveGAN | 0.0000 ± 0.0000 | 0.0003 ± 0.0006 | 0.0144 ± 0.0073 |
| FSDM | 0.4425 ± 0.0274 | 0.4446 ± 0.0413 | 0.4318 ± 0.0415 |
| SD | 0.3379 ± 0.0443 | 0.4359 ± 0.0172 | 0.4858 ± 0.0107 |
| SetGAN | 0.4979 ± 0.0361 | 0.531 ± 0.0355 | 0.5296 ± 0.0044 |
| Flowers | | | |
| AGE | 0.0033 ± 0.0029 | 0.0034 ± 0.0022 | 0.0180 ± 0.0046 |
| WaveGAN | 0.0000 ± 0.0000 | 0.0000 ± 0.000 | 0.0005 ± 0.0009 |
| FSDM | 0.1781 ± 0.0190 | 0.1789 ± 0.0178 | 0.1760 ± 0.0066 |
| SD | 0.2405 ± 0.0064 | 0.2863 ± 0.0154 | 0.3163 ± 0.0058 |
| SetGAN | 0.0217 ± 0.0070 | 0.0277 ± 0.0103 | 0.0276 ± 0.0026 |
| VGGFace | | | |
| AGE | 0.0098 ± 0.0043 | 0.0346 ± 0.0028 | 0.1103 ± 0.0035 |
| WaveGAN | 0.0000 ± 0.0000 | 0.0000 ± 0.0000 | 0.0014 ± 0.0009 |
| FSDM | 0.3631 ± 0.0035 | 0.3331 ± 0.0096 | 0.3082 ± 0.0259 |
| SD | 0.0624 ± 0.0049 | 0.0637 ± 0.0045 | 0.0674 ± 0.0044 |
| SetGAN | 0.5467 ± 0.0016 | 0.5602 ± 0.0029 | 0.5722 ± 0.0047 |

Table 9: F1 results on all datasets.

