# OpenReview forum: "Few Shot Image Generation Using Conditional Set-Based GANs"
_TMLR — Rejected by TMLR_

### Review · Reviewer_i9Wf · 2024-02-09

**Summary Of Contributions:**

The authors propose a method for image generation conditioned on a set of images. They focus on the task of few-shot image generation, where the model is given a few images from a class not seen during training (the conditioning set), and the goal is to generate novel images of that class. The authors claim that their model generates images that better capture the factors of variation within a(n unseen) class and with higher fidelity than previous methods. Further, they also discuss some of the limitations with the commonly used metrics to evaluate these few-shot methods and use MiFID as their main evaluation metric, which is a variation of FID scores that explicitly penalizes images that are too similar to seen examples.

**Audience:**

Yes

**Broader Impact Concerns:**

The authors have included a Broader Impact statement, which is appropriate considering this is a paper on image generation.

**Claims And Evidence:**

Yes

**Requested Changes:**

To properly assess whether the model captures the factor of variations in an unseen class better than baselines, I ask to conduct an experiment as follows:

1. Create classes with well-defined factors of variations - for example, dogs of a specific breed where all images have their ears pointing up, as done in some of the analysis.
2. Generate N (e.g. 100) samples for each model in the few-shot scenario described in the paper.
3. Train a model (or use human evaluation) to determine, for each generated example, whether i) it depicts the dog breed (dog breed classifier), and whether ii) the dog has or not the ears pointing up (ears classifier).
4. Count how many of the images actually belong to the class and how many fail the test because they do not.

This experiment can better assess whether the images stay in the class of the conditioning, and paired with image quality metrics can give a better overview of the performance of the model.

**Strengths And Weaknesses:**

Strengths:

[+] Clearly written

The paper is clearly written and presented, making it easy to follow along.

[+] Good experiment showing that FID clearly favors copies of the reference set.

The authors have conducted an experiment (mainly summarized in Table 2) to show that FID is not a good metric for their task, and to justify the use of MiFID. This is a convincing experiment and the conclusions support their hypothesis.

[+] Strong quantitative results

The results support that the model is state-of-the-art when using the MiFID metric and very competitive using LPIPS to competing baselines.


Weaknesses:

[-] Qualitative samples are not convincing

The authors claim that AGE generates images that are very close to the conditioning set and claim better generations from their model. However, the qualitative samples barely show this. In particular, I couldn't tell a big difference in neither Figure 3 or 4 between the generations of AGE and SetGAN, and actually some of the samples from SetGAN seem perceptually worse than those of AGE.

[-] Metrics are still inconclusive

The metrics used in the paper mainly evaluate image quality (LPIPS, MiFID), and implicitly some factors of variation by comparing populations (MiFID). However, image quality is only one part of what is expected out of the generations. The other aspect is whether the generations stay in-distribution in the class, but actually cover all possible factors of variation inside that class. I propose an experiment to analyze this quantitatively (not just qualitatively) in the section below.

[-] Minor: the captions of some of the figures do not match the text in the figures.

For example, in Figure 1 one of the model components is pixel2style2pixel without any abbreviation, while the captions mentions pSp. That abbreviation is not yet introduced in the text until a few pages later.

---

> ### Author Response · Authors · 2024-03-24
> **Response to Reviewer i9Wf**
>
> Thank you for your thorough and detailed review! We very much appreciate the detailed responses to our work. You raised a number of points in particular that we wish to focus on, and our responses to each point are listed below.
>
> 1) Qualitative results are unconvincing
>
> In your review, you state that the qualitative results in Figures 3 and 4 do not demonstrate superior generations for SetGAN compared to AGE. After examining the figures again, it is possible that the figures included in the paper are not high resolution enough to discern the differences. We have expanded both figures to attempt to make the distinctions between the two generations more clear. When examined closely, the AGE generations are often blurry and low-quality, and sometimes include significant distortions. In Figure 4, for example, consider all of AGE's generations for the AnimalFaces dataset. The dogs in the first row have blurry, distorted fur, and problematic features such as missing eyes (for the second image). All images in the third row are badly distorted and hardly recognizable as cats. Similarly, in Figure 3, the generations for Flowers and AnimalFaces are often washed out and blurry, and contain distortions that remove the key shared features of the inputs (such as the removed ears of the second dog, or the distortions in the second flower that have merged multiple flowers together). AGE's generations on VGGFace are typically higher quality, but still contain problematic elements such as removing the eye makeup in the third and fourth generated image of Figure 3.
>
> 2) Improved metrics for assessing class fidelity and coverage
>
> This is an excellent suggestion. Unfortunately, the suggested approach has several complications. The data used in the section on cross-class generation was hand-labelled by us, and labelling enough samples to train a classifier would be prohibitive.
>
> We do agree however about the importance of additional metrics to measure these qualities. Specifically, you mention in your review that there should be better ways to assess quantitatively whether the generations "stay in-distribution in the class, but actually cover all possible factors of variation inside that class". In order to add additional measures to assess this, we included Precision and Recall scores as suggested by Kynkäänniemi et al [1], as they are used to measure precisely these qualities. Table 1 in the paper has been reworked to include F1 score as the harmonic average of precision and recall for this purpose.
>
> [1] Kynkäänniemi et al., “Improved Precision and Recall Metric for Assessing Generative Models”.

---

### Review · Reviewer_ErYa · 2024-03-04

**Summary Of Contributions:**

In this paper, the authors study the problem of few-shot image generation using a set of unseen images. While prior works condition on a single image, authors propose to condition on reference sets of unseen classes. The proposed set-based GAN combines information from multiple reference images and generates diverse images within this new unseen reference class.  Finally, the authors identify limitations of existing evaluation metrics in few-shot image generation and discuss possible alternatives to overcome these limitations.

**Audience:**

No

**Broader Impact Concerns:**

The broader impact statement sufficiently addresses potential concerns.

**Claims And Evidence:**

No

**Requested Changes:**

### Critical Points


The points mentioned in the weakness section above are critical for securing a recommendation of acceptance.

### Good to incorporate

1. Table 2: MiFID numbers seem completely out-of-scale for the copied images. What might explain this behavior?

**Strengths And Weaknesses:**

###  Strengths


1. This paper studies the use of GANs in few-shot image generation task, which is an emerging area of research in the era of large-scale language-vision foundation models, such as Stable Diffusion.

2. The proposed method seems correct and sufficient experiments have been carried out to corroborate the main claims of the paper.

### Weaknesses

1. “While some limited equivalents exist for image-to-image generation such as inpainting Rombach et al. (2022) or image translation Saharia et al. (2022); Sasaki et al. (2021), no such large-equivalents exist for large-scale true few-shot generation of images conditioned on sets of unseen images.” Factual error: There exist large-scale few-shot generations of images conditioned on sets of unseen images. How does the proposed method compare with DreamBooth [1], HyperDreamBooth [2], IP-Adapter [3], LoRA [4] and many other follow-up works along this direction?


2. Simple extension of DAGAN (Antoniou et al., 2017) by newly added conditioning on multiple images.


3. Compared baselines are too weak compared to the related work cited in the paper, e.g. StableDiffusion based few-shot image generation.


4. Table 1 results show that the FID score decreases significantly whereas the LPIPS score increases by a slight margin. This essentially means the diversity is not quite achieved as claimed by the paper.


### References

[1] Ruiz et al., “DreamBooth: Fine Tuning Text-to-Image Diffusion Models for Subject-Driven Generation”.

[2] Ruiz et al., “HyperDreamBooth: HyperNetworks for Fast Personalization of Text-to-Image Models”.

[3] Ye et al., “IP-Adapter: Text Compatible Image Prompt Adapter for Text-to-Image Diffusion Models”

[4] Hu et al., “LoRA: Low-Rank Adaptation of Large Language Models” (adapted to vision: https://github.com/cloneofsimo/lora)

---

> ### Author Response · Authors · 2024-03-24
> **Response to Reviewer ErYa**
>
> Thank you very much for your review! You are correct that we did overlook some works that have succesfully adapted text-to-image models like Stable Diffusion to the domain of image-to-image generation. Upon reviewing the works you mentioned, we observe that all of these approaches continue to condition their generations on a single image, rather than a set. While these approaches have been highly successful, this may still be a limiting factor when generating conditioned on multiple images with shared features.
>
> In order to test this, we have added comparisons to the paper against a Stable Diffusion-based approach. We chose Stable Diffusion Image Variations (see [1] and [2]), as this was the easiest comparison to implement on a short timeframe. We found that SetGAN still outperformed the Stable Diffusion baseline on two out of three datasets under consideration - though the SD baseline did perform best on the low-data setting of the Flowers dataset, where its large-scale pretraining gave it a significant advantage over the other approaches. These results have been added to the tables in Section 5, as well as the qualitative comparisons in Figures 3 and 4.
>
> [1] https://www.justinpinkney.com/blog/2023/stable-diffusion-image-variations
>
> [2] https://huggingface.co/lambdalabs/sd-image-variations-diffusers
>
> As to your other points:
> 1) SetGAN does not achieve superior scores to the baselines in all settings for diversity.
>
> This is true. However, we would point out that the only baselines which achieve higher LPIPS scores are FSDM and Stable Diffusion. FSDM frequently has very poor class fidelity (as shown in Figures 3 and 4), which naturally increases the diversity of its generations. Stable Diffusion also leverages a much larger and more diverse training set, which also improves the diversity of its generations. We will be sure to update the wording in the paper to be more accurate about which settings it outperforms other models in terms of diversity and which it does not.
>
> 2) The large scale of MIFID scores for WaveGAN
>
> This is an artifact of the way MIFID scores are calculated, due to the very large multiplicative penalty that is introduced for models that exactly copy the inputs. While this is intended to some degree, we did consider other modifications of the MIFID score to keep the penalties on a more comparable scale. In the end, however, we decided to use the score as it was originally proposed in the interest of more standardized comparison.

---

### Review · Reviewer_Lfao · 2024-03-10

**Summary Of Contributions:**

This paper integrates transformer decoder blocks into GANs, facilitating multi-reference conditional generation, which enhances the diversity of few-shot generation outcomes. The authors conduct experiments and ablation studies to analyze the effectiveness of the proposed algorithm.

**Audience:**

Yes

**Broader Impact Concerns:**

No specific concerns were identified.

**Claims And Evidence:**

Yes

**Requested Changes:**

In addition to addressing the weaknesses, it's recommended to use high-resolution figures in Figure 1.

An intriguing extension of this work could involve conditioning on images from various classes, such as faces of people, dogs, and cats, to explore the model's generative capabilities further.

**Strengths And Weaknesses:**

Strengths:

The specifically designed architecture for conditional generation using multiple references without additional training or fine-tuning is commendable.

The thorough ablation studies investigating the effects of different modules contribute significantly to understanding the proposed algorithm.

Weaknesses:

The paper overlooks discussion on prompt-based algorithms for diffusion models, a burgeoning area for personalizing diffusion models with sets of images.

Considering the proposed network's reliance on a generation size of 1024 x 1024, the choice of 256 x 256 for generation may seem limited in scope.

While the paper claims better diversity compared to other approaches, it lacks explicit design discussion to promote diversity. It would be beneficial to delve into the influence of style vectors and permutation-equivariant attention. Besides, would it be better to add diversity penalty on the generation of a set of images?

---

> ### Author Response · Authors · 2024-03-24
> **Response to Reviewer Lfao**
>
> Thank you very much for your review! With regards to your specific questions and concerns:
>
> 1) Discussion of prompt-based diffusion models
>
> See our response to Reviewer ErYa
>
> 2) Generation size of 256
>
> This size was chosen as it was similar to the generation size of other comparable models such as AGE. The datasets on which we chose to compare are also largely 256x256 or 128x128, and thus larger output resolution was not a priority. Future work on broader datasets will likely use a larger generation size
>
> 3) Explicit design discussion to promote diversity
>
> Yes, this is a very interesting idea. Currently, our hope is that the model will learn to generate diverse sets in order to mimic the factors of variation in the reference set, as the discriminator can leverage the relationships between the different elements in order to discern whether the two sets are drawn from the same distribution. Adding an auxiliary objective to promote diversity is an interesting suggestion, however, and could certainly be included in the training scheme.

---

### Review · Reviewer_Ebbw · 2024-03-13

**Summary Of Contributions:**

The authors propose a method to perform few-shot generation with GANs conditioned on a small set of reference images. They achieve this using a set-based Generator and a set-based Discriminator that are constructed using Transformer blocks to achieve permutation equivariance. The authors compare with other GAN-based baselines and show performance improvements.

**Audience:**

Yes

**Broader Impact Concerns:**

No broader impact concerns.

**Claims And Evidence:**

No

**Requested Changes:**

I would like for the authors to include a discussion of ways to condition diffusion models on a small set of images and ideally compare their approach with some standard methods from this literature.

**Strengths And Weaknesses:**

Strengths:
* The paper is well-wrriten.
* The problem of few-shot generation with generative models is timely and I believe it is interesting to the TMLR audience.
* There are multiple ablation studies analyzing the effects of the various components of the method (e.g. conditioning network and base style vector).
* The paper improves the baseline methods.

Weaknesses:
* To me, the biggest weakness is that the authors ignore a large body of prior work in using diffusion models to do few-shot generation. Multiple works have been proposed to that end including LoRA, Dreambooth, Textual Inversion, Multiresolution Textual Inversion, etc. These works adapt a pre-trained network with minimal training so that it learns to generate images conditioned on a small set. A big advantage of these methods is that they can leverage extremely powerful, pre-trained models (such as Stable Diffusion) instead of training a new model. Newer versions of these techniques do not perform *any* training at all. Another way to condition diffusion models on a set of images is ControlNet.
* The aforementioned methods perform well and they are widely used in practice. Yet, the authors do not discuss such works and do not perform any comparisons with them. Hence, it is very hard to understand what's the power of the proposed framework and how useful will it be for researchers and practitioners.

---

> ### Author Response · Authors · 2024-03-24
> **Response to Reviewer Ebbw**
>
> Thank you very much for your review! As your concerns are very similar to those raised by Reviewer ErYa, see our response to their review for how we have altered the paper to accommodate your suggestions.

---

### Author Response · Authors · 2024-03-24
**Updated draft with Stable Diffusion baseline**

Thank you to all reviewers for your comments. We have updated the draft of the paper to include three major changes:

1) We have added a comparison to the pretrained Stable Diffusion Image Variations model from [HuggingFace](https://huggingface.co/lambdalabs/sd-image-variations-diffusers). We have updated Figure 3, Figure 4 and Table 1 to include these results.
2) We have added Precision, Recall and F1 scores as defined in Kynkäänniemi et al (2019). These have been incorporated into Table 1, as well as in the appendices.
3) We have updated the related work section with a brief discussion of the prompt-based models many of you have mentioned, including DreamBooth and IPAdapter.

All changes made to the paper in the revised draft have been highlighted in blue. Further details are contained in the individual responses to each reviewer.

---

### Decision · Action_Editor_XJiF · 2024-05-08

**Recommendation:** Reject

**Comment:**

All reviews are leaning towards rejection due to missing comparisons with several relevant and popular baselines in the diffusion model literature (including DreamBooth, HyperDreamBooth, IP-Adapter, LoRA, etc). Authors could consider including comparisons with these methods and resubmit the paper if they wish so.

**Audience:**

The problem of few-shot image generation is quite relevant and would be of wide interest to the community.

**Claims And Evidence:**

The paper proposes a GAN based method for few-shot generation by conditioning on a set of images. The reviewers have appreciated the ablation studies conducted in the paper to analyze the effect of the various components of the method. The problem of few-shot generation is also quite relevant and of wide interest to the community, however all reviewers agree that the paper isn't suitable for publication in the current form, primarily due to missing comparisons with several relevant and popular baselines in the diffusion model literature (including DreamBooth, HyperDreamBooth, IP-Adapter, LoRA, etc). Authors have included comparisons with a simple diffusion based baseline in the revision but the reviewers have found it to be insufficient to justify the claims in the paper.

**Resubmission Of Major Revision:**

The authors may consider submitting a major revision at a later time.